# Long Non-Coding RNA BNIP3 Inhibited the Proliferation of Bovine Intramuscular Preadipocytes via Cell Cycle

**DOI:** 10.3390/ijms24044234

**Published:** 2023-02-20

**Authors:** Wenzhen Zhang, Jianfang Wang, Bingzhi Li, Bing Sun, Shengchen Yu, Xiaoyu Wang, Linsen Zan

**Affiliations:** 1College of Animal Science and Technology, Northwest A&F University, Yangling 712100, China; 2National Beef Cattle Improvement Center, Northwest A&F University, Yangling 712100, China

**Keywords:** lncBNIP3, bovine intramuscular preadipocytes, proliferation, cell cycle, CDC6

## Abstract

The intramuscular fat (or marbling fat) content is an essential economic trait of beef cattle and improves the flavor and palatability of meat. Several studies have highlighted the correlation between long non-coding RNAs (lncRNAs) and intramuscular fat development; however, the precise molecular mechanism remains unknown. Previously, through a high-throughput sequencing analysis, we found a lncRNA and named it a long non-coding RNA BNIP3 (lncBNIP3). The 5′ RACE and 3′ RACE explored 1945 bp total length of lncBNIP3, including 1621 bp of 5′RACE, and 464 bp of 3′RACE. The nucleoplasmic separation and FISH results explored the nuclear localization of lncBNIP3. Moreover, the tissue expression of lncBNIP3 was higher in the longissimus dorsi muscle, followed by intramuscular fat. Furthermore, down-regulation of lncBNIP3 increased the 5-Ethynyl-2′- deoxyuridine (EdU)-EdU-positive cells. The flow cytometry results showed that the number of cells in the S phase was significantly higher in preadipocytes transfected with si-lncBNIP3 than in the control group (si-NC). Similarly, CCK8 results showed that the number of cells after transfection of si-lncBNIP3 was significantly higher than in the control group. In addition, the mRNA expressions of proliferative marker genes CyclinB1 (CCNB1) and Proliferating Cell Nuclear Antigen (PCNA) in the si-lncBNIP3 group were significantly higher than in the control group. The Western Blot (WB) results also showed that the protein expression level of PCNA transfection of si-lncBNIP3 was significantly higher than in the control group. Similarly, the enrichment of lncBNIP3 significantly decreased the EdU-positive cells in the bovine preadipocytes. The results of flow cytometry and CCK8 assay also showed that overexpression of lncBNIP3 inhibited the proliferation of bovine preadipocytes. In addition, the overexpression of lncBNIP3 significantly inhibited the mRNA expressions of CCNB1 and PCNA. The WB results showed that the overexpression of lncBNIP3 significantly inhibited the expression of the CCNB1 protein level. To further explore the mechanism of lncBNIP3 on the proliferation of intramuscular preadipocytes, RNA-seq was performed after interference with si-lncBNIP3, and 660 differentially expressed genes (DEGs) were found, including 417 up-regulated DEGs and 243 down-regulated DEGs. The KEGG pathway analysis showed that the cell cycle was the most significant pathway for the functional enrichment of DEGs, followed by the DNA replication pathway. The RT-qPCR quantified the expression of twenty DEGs in the cell cycle. Therefore, we speculated that lncBNIP3 regulated intramuscular preadipocyte proliferation through the cell cycle and DNA replication pathways. To further confirm this hypothesis, the cell cycle inhibitor Ara-C was used to inhibit DNA replication of the S phase in intramuscular preadipocytes. Herein, Ara-C and si-lncBNIP3 were simultaneously added to the preadipocytes, and the CCK8, flow cytometry, and EdU assays were performed. The results showed that the si-lncBNIP3 could rescue the inhibitory effect of Ara-C in the bovine preadipocyte proliferation. In addition, lncBNIP3 could bind to the promoter of cell division control protein 6 (CDC6), and down-regulation of lncBNIP3 promoted the transcription activity and the expression of CDC6. Therefore, the inhibitory effect of lncBNIP3 on cell proliferation might be understood through the cell cycle pathway and CDC6 expression. This study provided a valuable lncRNA with functional roles in intramuscular fat accumulation and revealed new strategies for improving beef quality.

## 1. Introduction

In farm animals, visceral and subcutaneous fat pads are considered wastes [1]. In contrast, intramuscular or marbling fat is essential for improving the flavor and palatability of meat, and has high commercial value, especially in beef [2,3]. Previous studies have proved that several factors affect intramuscular fat accumulation, such as breed, sex, age, and nutritional management level, of which breed is one of the most potent effects [4]. For example, Japanese Black (JB) cattle, one of the most popular breeds of Wagyu, is characterized by the ability to accumulate large amounts of intramuscular adipose tissue, far exceeding that of Qinchuan (QC) cattle, one of the main local breeds in China. At the cellular level, the contents of intramuscular fat are dependent upon intramuscular adipocyte hyperplasia (adipocyte number) and hypertrophy (adipocyte size) [5]. The hyperplasia of intramuscular adipocytes is significant because it provides sites for later marbling fat deposition [1]; the hyperplastic stage of intramuscular adipose is considered to develop from late pregnancy and continues well into postweaning [5]. Therefore, the study on the proliferation of intramuscular preadipocytes has a specific potential production value. In recent years, several studies have shown that lncRNAs are related to the proliferation of intramuscular preadipocytes.

One of the gene categories expressed by genomes through RNA polymerase II-mediated transcription, lncRNAs, are defined as transcripts of more than 200 nucleotides that are not translated into proteins and are often low in abundance and stability [6]. In the last decade, it has been increasingly demonstrated that many genomes, historically being regarded as junk DNA, are pervasively transcribed and produce numerous lncRNAs [7], many of which play essential functions in basic biological processes, including cell proliferation, differentiation, and apoptosis [8,9]. During the last few years, the genome of livestock has been intensively investigated in the context of lncRNA, including cattle, pigs, and chickens [10]. Subsequently, many studies related to the expression of lncRNA in livestock found that lncRNAs are closely related to a series of critical economic phenotypes, including meat production [11,12,13], meat quality [14,15,16], reproductive [17,18], and disease-resistant performances [19,20].

Recently, some studies have explored the potential regulation of lncRNA in the bovine intramuscular fat deposition through the whole-transcriptome landscape analysis [21,22]. However, the regulatory mechanisms of lncRNA on the intramuscular fat accumulation of bovine remains unclear. Therefore, it is essential to investigate the regulatory mechanisms of lncRNAs on the proliferation of bovine intramuscular preadipocytes.

The accurate transition of the cell cycle at different phases is crucial for controlling eukaryotic cell proliferation, especially from G1 to the S phase [23]. The S phase of each cell cycle is achieved by duplication of chromatin structure in tight coordination with DNA replication; therefore, accurate DNA replication is essential for genome stability and cell viability [24,25]. Cell division cycle 6 (CDC6) is initially identified in a genetic screen, and its accumulation promotes the initiation of DNA replication and cell proliferation [26,27]. The Ara-C affects the molecule’s rotation within DNA and inhibits DNA polymerase, decreasing DNA replication and repair.

We found differentially expressed lncBNIP3 in the intramuscular fat of QC and JB cattle by high-throughput sequencing (unpublished) and speculated that lncBNIP3 could regulate intramuscular adipogenesis. In this study, we explored the effect and mechanism of lncBNIP3 on intramuscular adipocyte proliferation, as it might be beneficial for the genetic improvement of cattle meat quality traits.

## 2. Results

### 2.1. The Biological Characteristics and Expression Profile Analysis of lncBNIP3

According to the sequencing data, lncBNIP3 was located on chromosome 26, and there were four annotated genes near its 100 Kbp, among which BNIP3 was the closest (Figure 1A). The total length of lncBNIP3 amplified by the RACE technique was 1945 bp, with the amplification length of 5′RACE being 1621 bp and that of 3′RACE being 464 bp (Figure 1B). The coding potential of lncBNIP3 was predicted to be 0.049 and 0.107 by CPC2 and CPAT, respectively (Figure 1C). However, the mRNA coding potential of BNIP3 in positive control was all above 0.99 (Figure 1C). These results indicated that lncBNIP3 had little coding potential. Nucleoplasmic separation and FISH results indicated that lncBNIP3 was localized in the nucleus (Figure 1D–F). The transcriptome sequencing and quantitative results of lncBNIP3 expression were similar. Moreover, the expression of lncBNIP3 in the intramuscular fat of QC cattle was about twice as much as that of JB cattle (Figure 1G). The tissue expression profile showed that lncBNIP3 was most highly expressed in the longissimus muscle, followed by intramuscular fat among the eight tissues of the two cattle breeds (Figure 1H). During the proliferation of intramuscular preadipocytes, the expression of lncBNIP3 showed an increasing trend overall (Figure 1I).

### 2.2. Interference with lncBNIP3 Promoted the Proliferation of Bovine Intramuscular Preadipocytes

To investigate the effect of lncBNIP3 on the proliferation of bovine intramuscular preadipocytes, si-lncBNIP3 was synthesized and transfected into the bovine intramuscular preadipocytes. We found that the expression of lncBNIP3 was significantly down-regulated after transfection of si-lncBNIP3 for 24 h and 48 h (Figure 2A). The EdU results showed that the positive cell rate after transfection of si-lncBNIP3 for 48 h was significantly higher than that of the control group (Figure 2B,C). The flow cytometry results showed that the number of cells in the S phase after transfection of si-lncBNIP3 for 48 h was significantly higher than in the control group (Figure 2D,E). Similarly, CCK8 results showed that the number of cells after transfection of si-lncBNIP3 was significantly higher than that after transfection of si-NC for 48 h (Figure 2F). In addition, the mRNA expressions of proliferative marker genes CCNB1 and PCNA in the si-lncBNIP3 group were significantly higher than in the control group (Figure 2G). Furthermore, the Western Blot results showed that the protein expression level of PCNA after transfection of si-lncBNIP3 was significantly higher than in the control group, while there was no significant change in the expression level of the CCNB1 protein (Figure 2H,I). These results demonstrated that interference with the expression of lncBNIP3 had a positive effect on the proliferation ability of bovine adipocytes.

### 2.3. Overexpression of lncBNIP3 Inhibited the Proliferation of Bovine Intramuscular Preadipocytes

The overexpressed lncBNIP3 plasmid (OE-lncBNIP3) was designed and transfected into the bovine intramuscular preadipocytes. The results showed that lncBNIP3 was overexpressed nearly 70 times after transfection for 48 h (Figure 3A). The EdU results showed that the positive-cell rate was significantly lower after overexpression of lncBNIP3 for 48 h (Figure 3B,C). The flow cytometry results showed that overexpression of lncBNIP3 in the bovine intramuscular preadipocytes decreased the percentage of cells in the S phase, but the results were statistically insignificant (*p* = 0.084) (Figure 3D,E). The CCK8 results showed that the overexpression of lncBNIP3 inhibited the proliferation of bovine intramuscular preadipocytes (Figure 3F). In addition, the overexpression of lncBNIP3 significantly inhibited the mRNA expressions of CCNB1 and PCNA (Figure 3G). The Western Blot results showed that overexpression of lncBNIP3 significantly inhibited the expression of CCNB1 protein level, while there was no significant change in the PCNA protein level (Figure 3H,I). These results demonstrated that the overexpression of lncBNIP3 had a negative effect on the proliferation ability of bovine adipocytes.

### 2.4. Identification and Enrichment Analyses of DEGs Associated with lncBNIP3

To further explore the lncBNIP3 mechanism on the proliferation of intramuscular preadipocytes, RNA-seq was performed after interference with lncBNIP3 for 48 h. First, the results of Principal Component Analysis (PCA) and Pearson Correlation Coefficient analysis between samples showed that the control group and the treatment group were grouped into two separate clusters (Figure 4A,B), indicating that biological duplication met the requirements of subsequent analysis. Next, we analyzed gene expression differences between groups, and a total of 660 differentially expressed genes (DEGs) satisfying the conditions (*P*adj < 0.05 and log2 (fold change) > 1) were obtained, including 417 up-regulated DEGs and 243 down-regulated DEGs (Figure 4C). In addition, the heatmap analysis was conducted according to the expression of DEGs (Figure 4D), and the gene expression difference multiples and q-value between the two groups were shown (Figure 4E). After that, the functional enrichment analysis of DEGs was performed. For the biological process, GO analysis showed that the number of enriched genes in multiple pathways of the biological process was similar. They concentrated on the regulation of transcription and DNA-templated processes. For cellular components, most DEGs were enriched in the nucleus. For molecular function, protein binding had the most enriched genes (Figure 4F). The KEGG pathway analysis showed that the cell cycle was the most significant pathway for the functional enrichment of DEGs, followed by the DNA replication pathway (Figure 4G). Among them, 22 DEGs were in the cell cycle pathway, and 21 of them were up-regulated; in the DNA replication pathway, all 12 DEGs were up-regulated (Figure 4H).

### 2.5. Interference with lncBNIP3 Promoted the Cell Cycle and Rescued the Defective Cell Cycle Caused by Ara-C

The expression of some genes enriched in cell cycle and DNA replication pathways were used for RT-qPCR validation, and the results confirmed the consistency between RNA-seq and RT-qPCR (Figure 5A–D). Therefore, we speculated that lncBNIP3 has a regulation role in intramuscular preadipocyte proliferation through the cell cycle and DNA replication pathways. To further confirm this hypothesis, intramuscular preadipocytes were added with cell cycle inhibitor Ara-C, which inhibits DNA replication during the S phase of the cell cycle. Herein, different concentrations of Ara-C were added to intramuscular preadipocytes, and the CCK8 was performed. The results showed that the cells’ proliferation ability decreased gradually with the increase of Ara-C concentration, and the inhibitory effect was already significant when the concentration was at 0.25 μM. Therefore, cells used in this study were treated with a 0.25 μM concentration of Ara-C for subsequent experiments (Figure 5E). Bovine intramuscular preadipocytes were treated with Ara-C and si-lncBNIP3 simultaneously, and the CCK8 results showed that si-lncBNIP3 could rescue the inhibitory effect of Ara-C on the bovine intramuscular preadipocyte proliferation (Figure 5F). Meanwhile, the flow cytometry results showed that the bovine intramuscular preadipocytes kept getting stuck in the S phase by Ara-C, while si-lncBNIP3 could rescue this process (Figure 5G,H). Compared with the control group, the rate of EdU-positive cells increased significantly in Ara-C, Ara-C + silncBNIP3, and si-lncBNIP3 groups. However, the nuclear integrity of EdU-positive cells was inhibited by Ara-C, and the effect of Ara-C was rescued by si-lncBNIP3 (Figure 5I,J). The above results suggested that si-lncBNIP3 promoted the cell cycle and rescued the defective cell cycle caused by Ara-C.

### 2.6. Interference with lncBNIP3 Promoted the Promoter Activity and Expression of CDC6

To explore the genes directly regulated by lncBNIP3 in inhibiting cell proliferation, several biotin probes of lncBNIP3 were synthesized, and the DNA bound to it was pulled down. Subsequently, the DNA obtained above was fragmented by sonication, with most fragments ranging from 250 bp to 750 bp (Figure 6A). A total of 1027 peaks located in the promoter region were obtained by sequencing. An amount of 7 peaks were measured by RT-qPCR randomly, and the results showed that these peaks did exist and had a certain abundance in the products after fragmentation (Figure 6B). There were 34 overlapping genes between these peaks’ corresponding genes and DEGs of RNA-seq (Figure 6C). In functional enrichment analysis, these shared genes were most significantly enriched in the nucleus followed by cell division (Figure 6D). Both pathways included the CDC6, a proliferation-related gene, and RT-qPCR results confirmed the presence of the CDC6 promoter region in the products after fragmentation (Figure 6E). Fluorescence activity assay results showed that interference with lncBNIP3 promoted the promoter activity of CDC6 in intramuscular preadipocytes (Figure 6F). In addition, RNA-seq and RT-qPCR analysis showed that the mRNA expression of CDC6 was increased by interfering with lncBNIP3 (Figure 6G). Meanwhile, the Western Blot results showed that interference with lncBNIP3 significantly promoted the expression of CDC6 protein (Figure 6H,I). These results suggested that the effect of lncBNIP3 on the cell proliferation cycle can be understood by CDC6 expression.

### 2.7. LncBNIP3 Had No Significant Effect on the Apoptosis of Bovine Intramuscular Preadipocytes

To investigate whether the effect of lncBNIP3 on the proliferation of intramuscular preadipocytes is related to apoptosis, the Annexin V-FITC Apoptosis Detection Kit was used to analyze apoptosis differences between si-NC and si-lncBNIP3. The results of the flow cytometry showed that interference with lncBNIP3 had no significant effect on apoptosis (Figure 7A,B). Meanwhile, the mRNA expressions of Bax and Caspase3 had no significant change after transference with si-lncBNIP3 and si-NC (Figure 7C). Further, the same analysis was performed after overexpressing lncBNIP3, and the results of flow cytometry showed that there was still no significant effect on apoptosis (Figure 7D,E). Additionally, there was also an insignificant change in the mRNA expression of Bax and Caspase3 (Figure 7F). These results indicated that the effect of lncBNIP3 on cell proliferation was not affected by apoptosis.

## 3. Discussion

In recent years, lncRNAs have attracted much attention, as the number and types of known functional lncRNAs increased considerably and were known to be involved in the cis-regulation of target gene loci or near the same genomic locus [28]. Many studies have shown that lncRNAs lacking protein-coding potential tend to be more localized in the nucleus than mRNAs [29], influencing different biological processes by regulating the nuclear structure, the transcription in the nucleus, and mRNA stability, translation, and post-translational modification in the cytoplasm [30]. Intramuscular fat (IMF) content is the best index of beef grade and plays a crucial role in beef flavor [31,32]. Some lncRNAs, whose expression was altered during the proliferation and differentiation of intramuscular preadipocytes, had been demonstrated to regulate intramuscular adipogenesis [33,34]. In this study, lncBNIP3 was found and located near BNIP3 (Figure 1A), a BH-3-only Bcl-2 family member which could induce cell apoptosis [35]. However, lncBNIP3, unlike BNIP3, had no significant effect on cell apoptosis (Figure 7). LncBNIP3 was localized in the nucleus (Figure 1D–F) so it was speculated that lncBNIP3 could be an important regulator of nuclear organization and function [30]. Indeed, subsequent studies demonstrated that the regulation of lncBNIP3 in the cell cycle did occur in the nucleus (Figure 5). In the present study, the expression of lncBNIP3 showed a significant increasing trend during the proliferation of bovine intramuscular preadipocyte (Figure 1I), and lncBNIP3 was highly expressed in intramuscular fat when compared with other tissues (Figure 1H). These results primarily showed that lncBNIP3 could affect intramuscular adipocyte proliferation. Moreover, the lncBNIP3 expression level was higher in the longissimus muscle when compared with the other seven tissues (Figure 1H), which suggested that lncBNIP3 could play a crucial role in myogenesis. Intramuscular fat deposition varied significantly among different varieties. To analyze whether lncBNIP3 was really involved in the production of intramuscular fat in cattle, we selected the famous Chinese breed (QC cattle) and the world-famous breed with an intramuscular adipose deposit (JB cattle) for comparative analysis of lncBNIP3, and the results showed that the expression of lncBNIP3 in the intramuscular fat tissue of JB cattle was significantly lower than in QC cattle (Figure 1G). Therefore, the functional study of lncBNIP3 was meaningful for livestock production.

Our study found that interference with lncBNIP3 promoted the proliferation of bovine intramuscular preadipocytes (Figure 2). In contrast, the overexpression of lncBNIP3 inhibited the proliferation of bovine intramuscular preadipocytes (Figure 3). To further identify the regulatory factors of lncBNIP3 during lipid deposition in the bovine preadipocytes, RNA-seq was used to analyze the regulatory mechanism of lncBNIP3 on cell proliferation. KEGG pathway analysis demonstrated that lncBNIP3 influenced the cell cycle pathway and DNA replication (Figure 4). The cell cycle represented a series of tightly integrated events to determine cell growth and proliferation [36]. Herein, 21 DEGs were found to be related to the cell cycle, including CDCs, CCNB1, MCMs, CCNA2, and other marker genes. The result of RT-qPCR further confirmed that si-lncBNIP3 significantly promoted its expression (Figure 5B). Moreover, complete and accurate DNA replication is fundamental to cellular proliferation and genome stability [37]. Six genes related to this pathway were also quantitatively analyzed, which confirmed the reliability of the sequencing results.

Ara-c is a cytosine antimetabolite capable of interfering with cell proliferation by inhibiting DNA synthesis. The defective cell cycle and proliferation caused by Ara-C were rescued by interference with lncBNIP3 (Figure 5F–J), ceasing the S phase of the cell cycle and a decrease in DNA replication [38]. As a crucial replication licensing factor, CDC6 was pivotal in the regulation of the DNA replication process and cell proliferation [27,39]. A study has shown that CDC6 is involved in the regulation of the S and M phases of the cell cycle in eukaryotic cells [40]. Inhibition of CDC6 expression is beneficial for G1/S phase stagnation [32]. The lncRNA’s regulation of CDC6 has also been reported (such as lncrNa-CDC6); however, we found that it was mainly concentrated in cancer-related studies, and no relevant report was found in adipocyte development [41]. Herein, we found that lncBNIP3 could bind directly to the promoter of CDC6, and the expression of CDC6 was increased when interfered with lncBNIP3 (Figure 6). These results suggested that lncBNIP3 suppressed the proliferation of intramuscular preadipocytes by acting on CDC6; however, why and how lncBNIP3 could bind CDC6 remained unclear and needed further exploration. Additionally, we also attempted to explore the role of lncBNIP3 in apoptosis. Strikingly, the results of flow cytometry and expression of apoptosis marker genes exhibited a non-significant effect of lncBNIP3 in preadipocyte apoptosis. Our study suggested that lncBNIP3 is crucially involved in the intramuscular fat accumulation of beef cattle, which is beneficial for the genetic improvement of cattle meat quality traits.

## 4. Materials and Methods

### 4.1. Animal, Samples and Cell Culture

In this study, three QC cattle and three JB cattle of the same age (3 years old), gender (male), and similar management conditions were selected from National Beef Cattle Improvement Research Center. The animals were fed a total mixed ration (TMR), containing 25% concentrate and 75% roughages of dry straw and corn silage; water was offered ad libitum. The feeding was offered based on NRC standards (Nutrient Requirement of Beef Cattle) and in a similar rearing environment (similar temperature, humidity, etc.). The animals were slaughtered, and their eight tissues were isolated, including the spleen, liver, lung, kidney, heart, subcutaneous fat, longissimus muscle, and intramuscular fat. The isolation and culture of bovine intramuscular preadipocytes were the same as described previously [22,42]. The isolated cells were cultured in a 5% CO_2_ and 37 °C incubator with DMM/F12 (Gibco, Grand Island, NY, USA) containing 10% FBS (Invitrogen, Invitrogen, Waltham, MA, USA) and 1% antibiotics (100 IU/mL penicillin and 100 μg/mL streptomycin).

### 4.2. Fragment Synthesis and Recombinant Vector Construction

The full-length fragment of lncBNIP3 was synthesized and subcloned into the pcDNA3.1 vector to construct the overexpression vector of lncBNIP3, named OE-lncBNIP3. The promoter fragment of CDC6 was synthesized, sequenced, and then subcloned into a pGL3 vector, named CDC6-p. The small interfering RNA (siRNA) of lncBNIP3 was designed and synthesized by GenePharma (Shanghai, China) (siRNA sequence: 5′- CUUGUGUCCUGCACUUUGATT-3′), named si-lncBNIP3.

### 4.3. Cell Transfection

After growth to 50–60% confluence, the intramuscular preadipocytes were transfected with siRNA (50 nM), OE-lncBNIP3 (1250 ng/mL) using Lipofectamine 3000 reagent (Thermo Fisher Scientific, Waltham, MA, USA) according to the manufacturer’s instructions, respectively. Briefly, RNA oligonucleotides and the transfection reagent were separately diluted in Opti-MEM medium (Gibco) and incubated for 10 min at room temperature. Next, the two mixtures were combined and incubated for another 15 min at room temperature to allow the formation of transfection reagent-RNA complexes. The transfection complexes were then added to the cell culture medium dropwise. The cells were incubated for 24 h before changing to a fresh medium.

### 4.4. 5′ and 3′ Rapid Amplification of cDNA Ends (RACE)

5′ RACE and 3′ RACE were performed using the SMARTer RACE 5′/3′ Kit (Takara, Beijing, China) according to the manufacturer’s instructions. The following gene-specific primers (GSP) are used for PCR: 5′-GATTACGCCAAGCTTTCCACCTTCCCCCTGGTTGTTTATC-3′ (5′ RACE GSP1), 5′-GATTACGCCAAGCTTTGCTCTCTGGTCCGCTCTGGTTA-3′ (3′ RACE GSP1).

### 4.5. Cytoplasmic and Nuclear RNA Separation

Cytoplasmic and nuclear RNA separation from intramuscular preadipocytes was performed using the PARIS^TM^ Kit (Life Technologies Corporation, Carlsbad, CA, USA) according to the manufacturer’s instructions. Briefly, 10^7^ cells were harvested and then incubated in the Cell Disruption Buffer for 10 min. The cytomembrane of the cells was cleaved. Subsequently, the cells were separated into a cytoplasmic fraction and a nuclear fraction. Finally, the RNA of the cytoplasmic and nuclear fractions was isolated using Ambion RNAqueous^®^ technology.

### 4.6. Fluorescence In Situ Hybridization (FISH)

According to the manufacturer’s protocol, the subcellular localization of lncBNIP3 was detected by Ribo^TM^ Fluorescent in Situ Hybridization Kit (Ribobio, Guangzhou, China). The Cy3-labeled lncBNIP3 probes were designed and synthesized by Ribobio. The images were captured using a microscope (Olympus, Shinjuku City, Japan).

### 4.7. Cell Proliferation and Apoptosis Analysis

Cell proliferation was measured using the Cell Counting Kit-8 (CCK8) (Beyotime, Shanghai, China), 5-Ethynyl-2-deoxyuridine (EdU) assay, and cell cycle assay. The CCK8 assay was performed using a CCK8 kit (Bioscience, Shanghai, China) according to the manufacturer’s instructions with a microplate reader (Infinite M200 PRO; Tecan, Switzerland). EdU assays were performed using a Cell-Light EdU DNA proliferation kit (RiboBio, Guangzhou, China) according to the manufacturer’s instructions with EVOSTM Auto 2 (ThermoFisher, Waltham, MA, USA). Cell cycle assay was performed with the Cell Cycle Straining Kit (MultiSciences, Hangzhou, China) according to the manufacturer’s instructions with ModFit LT^TM^ (version 5.0; Verity Software House, Topsham, ME, USA). Cell apoptosis was measured using Annexin V-FITC Apoptosis Detection Kit according to the manufacturer’s instructions with ModFit LT^TM^ (version 5.0; Verity Software House, USA).

### 4.8. Real-Time Quantitative PCR (RT-qPCR)

The total RNA was extracted from the tissue using the TRIzol™ Reagent (Invitrogen, Thermo Fisher Scientific, Inc. USA). The integrity (quantity and quality) of the total extracted RNA was checked through an optical density of 260 and the ratio of the optical density (OD) of 260/280 using the Nano Quant plate^TM^ (TECAN, Infinite M200 PRO), and was further verified through 1% agarose gel. The cDNA libraries were constructed using a PrimeScriptTM RT reagent kit with a gDNA eraser (Perfect Real Time, Takara). RT-qPCR was performed following the manufacturer’s protocol of Sybr Premix EX Taq Kit (Takara, Dalian, China) using thermocycler 7500 system SDS V 1.4.0 (Applied Biosystem, Waltham, MA, USA). The bovine *β-Actin* gene was used as an endogenous control. The thermocycling conditions were as follows: pre-heating at 95 °C for 5 min, a total of 34 cycles of denaturation at 95 °C for 30 s, annealing temperature at 60 °C for 30 s, and extension temperature at 72 °C for 30 s. The relative mRNA expression levels were calculated using the 2^−ΔΔCt^ method.

### 4.9. Western Blot Analysis

Western blot was performed as previously described [43]. Four primary antibodies (CCNB1 [55,004-1-AP, 1:500, Proteintech], PCNA [10,205-2-AP, 1:2000; Proteintech, Rosemont, USA], β-actin [6008-1-Ig, 1:5000, Proteintech], and CDC6 [D160157, 1:500, Sangon Biotech, Shanghai, China]), and two secondary antibodies (ab150113 and ab150077, 1:5000, Abcam, Cambridge, UK) were used in this study. Western blot quantification was measured using Image Lab (Bio-Rad, Hercules, CA, USA).

### 4.10. RNA Isolation, Library Construction, and Sequencing

Total RNA was extracted using the Trizol reagent (Invitrogen, USA) with the manufacturer’s instructions to further detect its concentration and integrity. Three µg of RNA from each qualified sample was sent to Lc-Bio Technologies Co., LTD (Hangzhou, China) for library construction. The library construction mainly included the following steps: rRNAs were removed to obtain mRNAs, PCR amplified and sequenced using Illumina PE150; raw image files obtained from high-throughput sequencing were converted into sequenced reads identified by CASAVA bases and stored in FASTQ format.

### 4.11. Raw Data Quality Control and Data Analysis

Raw data were filtered from the adapters, and low-quality fragments were sequenced using Trimmomatic to obtain high-quality clean reads for further downstream analyses [44]. FastQC (https://www.bioinformatics.babraham.ac.uk/projects/fastqc/, accessed on 15 May 2022) was used to check the quality of clean reads, including Q20, Q30, and GC content. Furthermore, the clean reads were mapped to the *Bos taurus* (ARS-UCD1.2) reference genome using HISAT2 [45]. StringTie was used to assemble and quantify mapped reads [46]. Cuffmerge was used to merge the transcripts spliced from the samples. For significantly different expressions, edgeR [47] was used to perform analysis with corrected *p*-value (*P*adj) < 0.05 and log2 (fold change) > 1. The GOseq R package implemented gene ontology (GO) enrichment analysis [48]. KOBAS was used to perform the detected KEGG pathway analysis [49]. *P*adj < 0.05 was selected as the reliability screening criterion.

### 4.12. Chromatin Isolation by RNA Purification (ChIRP)

The ChIRP assay was performed using the Chromatin Isolation by RNA Purification (ChIRP) Kit (BersinBio, Guangzhou, China) following the manufacturer’s instructions. Subsequently, ChIRP-seq and the combined DNA analysis were performed by Yimai (Guangzhou) Biomedical Technology Co., LTD (Guangzhou, China).

### 4.13. Luciferase Reporter Assays

The intramuscular preadipocytes were cultured in 24-well plates. When cell fusion reached 80–90%, the CDC6-p vector was co-transfected with si-lncBNIP3 or siRNA negative control (si-NC) using Lipofectamine™ 3000 reagent, respectively. After 48 h of transfection, firefly luciferase activities were measured using the Dual-Luciferase Reporter Assay System (Promega, Madison, WI, USA) according to the manufacturer’s instructions.

### 4.14. Statistical Analysis

One-way ANOVA analysis and two-way ANOVA were used for multiple groups, and two-group comparisons were analyzed via the *t*-test by using IBM SPSS Statistics software (version 26.0; IBM Corp., Armonk, NY, USA), and expressed as the Mean ± Standard Deviation (n = 3). Group differences were considered statistically significant at *p* < 0.05 or *p* < 0.01, and different lower cases among different columns represent *p* < 0.05. GraphPad Prism 6.01 (San Diego, CA, USA) was used to generate the graphs. The visualization of RNA-seq analysis was completed in the R language environment.

## 5. Conclusions

Summarily, lncBNIP3, which was differentially expressed in the intramuscular fat of QC cattle and JB cattle, inhibited the proliferation of bovine intramuscular preadipocytes. Mechanistically, the inhibitory effect of lncBNIP3 on the proliferation of bovine intramuscular preadipocytes might be understood through the regulation of the cell cycle, DNA replication pathways, and the direct regulation of CDC6 expression (Figure 8).

## Figures and Tables

**Figure 1 ijms-24-04234-f001:**
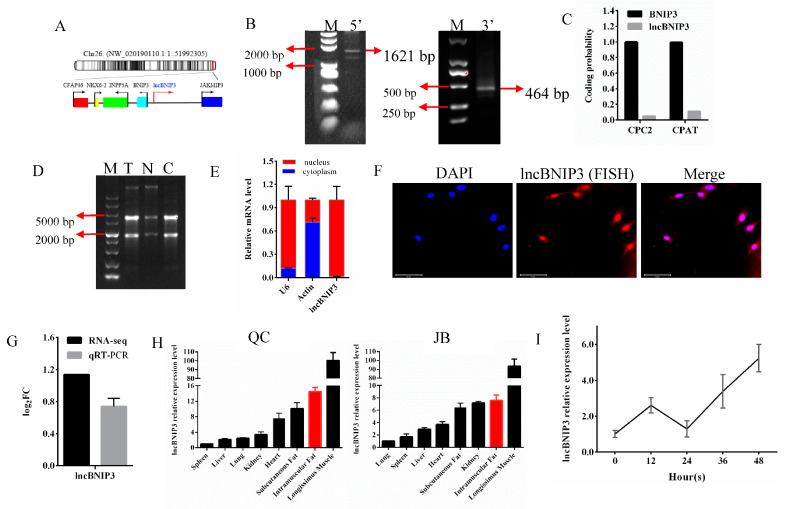
The biological characteristics and expression profile analysis of lncBNIP3. (**A**) Location of lncBNIP3 on chromosomes; (**B**) Gel electrophoresis results of RACE product (M for the marker, a fragment size of 5′RACE gel marker, from top to bottom, was 5000 bp, 3000 bp, 2000 bp, 1000 bp, 750 bp, 500 bp, 250 bp, 100 bp; a fragment size of 3′RACE gel marker, from top to bottom, was 2000 bp, 1000 bp, 750 bp, 500 bp, 250 bp, 100 bp); (**C**) Prediction results of lncBNIP3 encoding potential in CPC2 and CPAT, and BNIP3 mRNA was a positive control; (**D**) Gel electrophoresis of cytoplasmic and nuclear RNA separation (M for marker, T for total RNA, N for nuclear RNA, C for cytoplasmic RNA); (**E**) The expression of lncBNIP3 in nucleus and cytoplasm (U6 and β-actin were controls); (**F**) FISH results of lncBNIP3, scale bar: 200 μm; (**G**) The expression levels of lncBNIP3 in QC cattle and JB cattle were detected by RNA-seq and RT-qPCR. (**H**) The tissue expression profiles of QC Cattle and JB Cattle; (**I**) The expression of lncBNIP3 during intramuscular preadipocyte proliferation.

**Figure 2 ijms-24-04234-f002:**
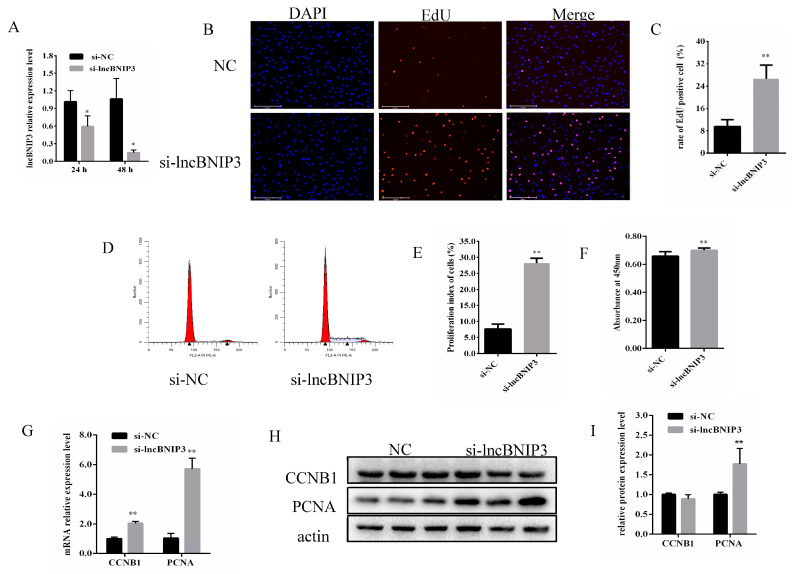
Interference with lncBNIP3 promoted the proliferation of bovine intramuscular preadipocytes. (**A**) Interference efficiency of si-lncBNIP3 for 24 h and 48 h; (**B**) EdU staining results (200×), scale bar: 200 μm; (**C**) Statistical data of the EdU results; (**D**) Flow cytometry results; (**E**) Statistical data of the flow cytometry results; (**F**) Statistical data of the CCK8 results; (**G**) The mRNA expression of CCNB1 and PCNA; (**H**) Western Blot results of CCNB1 and PCNA; (**I**) Statistical data of the Western Blot results. Data are expressed as means ± SD. ** indicated extremely significant difference (*p* < 0.01), * indicated significant difference (*p* < 0.05).

**Figure 3 ijms-24-04234-f003:**
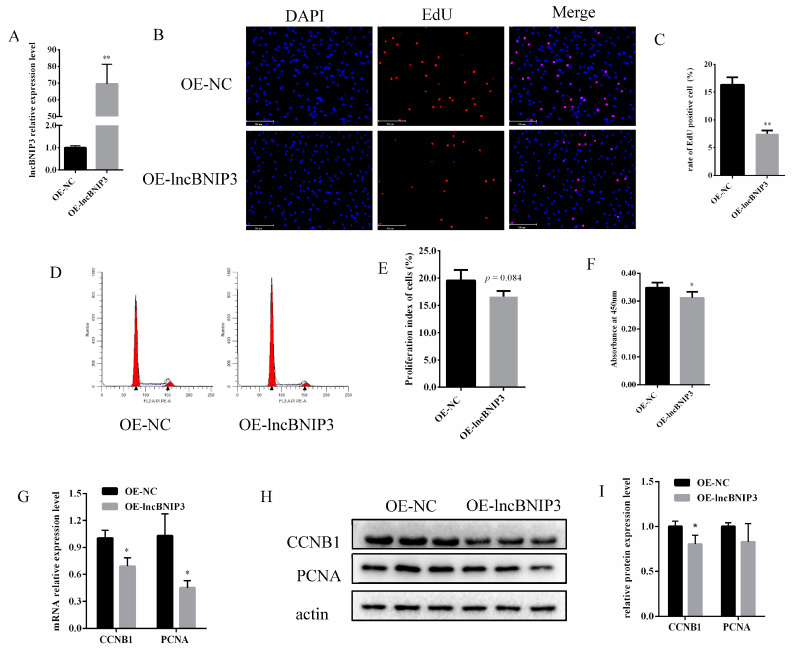
Overexpression of lncBNIP3 inhibited the proliferation of bovine intramuscular preadipocytes. (**A**) The efficiency of lncBNIP3 overexpression; (**B**) The EdU staining results (200×), scale bar: 200 μm; (**C**) Statistical data of the EdU results; (**D**) Flow cytometry results; (**E**) Statistical data of the flow cytometry results; (**F**) The results of CCK8; (**G**) The expression levels of CCNB1 and PCNA in mRNA levels; (**H**) Western Blot results of CCNB1 and PCNA; (**I**) Statistical data of the Western Blot results. Data are expressed as means ± SD. ** indicated extremely significant difference (*p* < 0.01), * indicated significant difference (*p* < 0.05).

**Figure 4 ijms-24-04234-f004:**
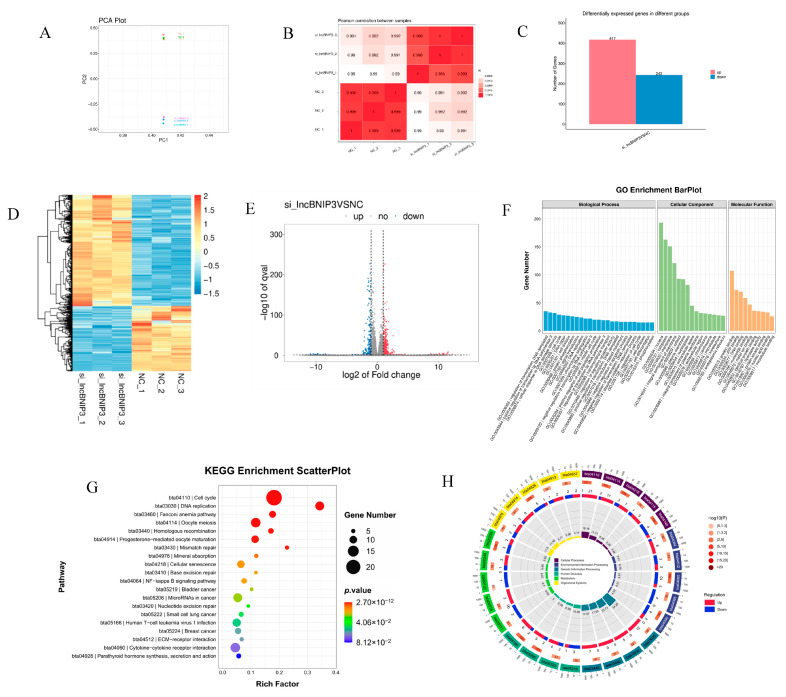
Identification and enrichment analyses of DEGs associated with lncBNIP3. (**A**) PCA Plot; (**B**) Pearson correlation coefficient cluster graph; (**C**) The number of up-regulated and down-regulated DEGs; (**D**) Heatmap for hierarchical cluster analysis of all sequencing samples and DEGs; (**E**) Volcano plot for the distribution of DEGs; (**F**) GO enrichment analysis results for DEGs; (**G**) KEGG enrichment analysis of DEGs; (**H**) Cyclic plots of the number of up-regulated and down-regulated DEGs in different enrichment pathways.

**Figure 5 ijms-24-04234-f005:**
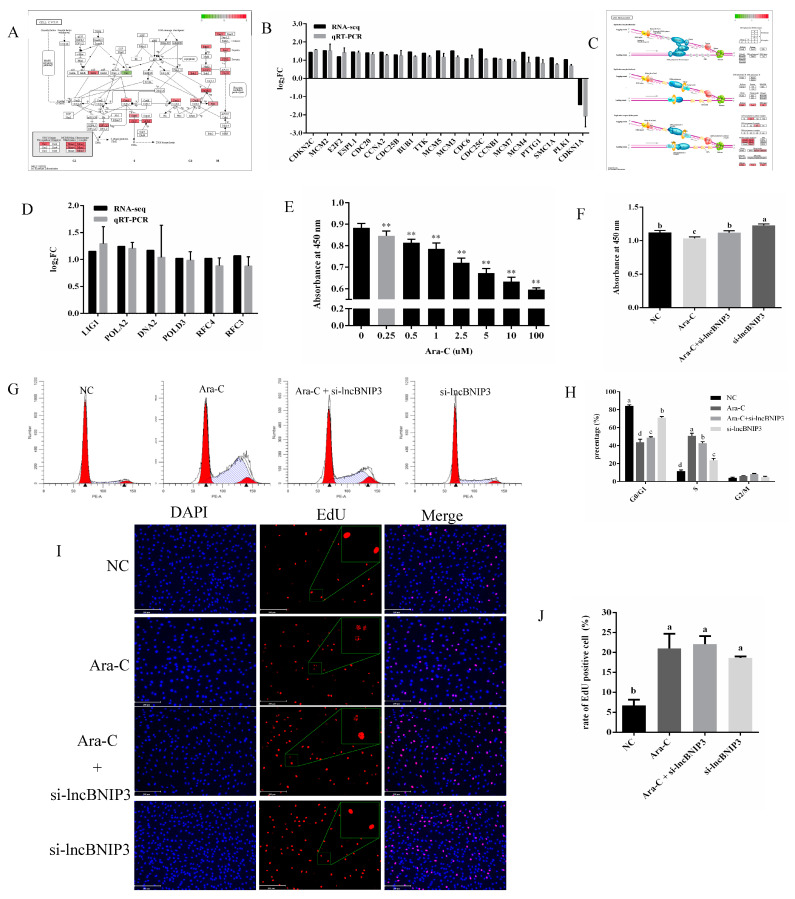
Interference with lncBNIP3 promoted the cell cycle and rescued the defective cell cycle caused by Ara-C. (**A**) KEGG pathway for cell cycle (bta04110) with the annotated DEGs. (**B**) Sequencing data approved by RT-qPCR. (**C**) KEGG pathway for DNA replication (bta03030) with the annotated DEGs. (**D**) Sequencing data approved by RT-qPCR. (**E**) The CCK8 results of cells treated with Ara-C at different concentrations. (**F**) The CCK8 results of NC, Ara-C, Ara-C + si-lncBNIP3, and si-lncBNIP3 groups. (**G**) Cell flow cytometry results of NC, Ara-C, Ara-C + si-lncBNIP3, and si-lncBNIP3 groups (**H**) Statistical data of the flow cytometry results. (**I**) The EdU staining results of NC, Ara-C, Ara-C + si-lncBNIP3, and si-lncBNIP3 groups, scale bar: 200 μm. (**J**) Statistical data of EdU-positive cell rate. Herein, Data are expressed as means ± SD. ** indicated extremely significant difference (*p* < 0.01). And different lower cases (a–d) among different columns represent *p* < 0.05.

**Figure 6 ijms-24-04234-f006:**
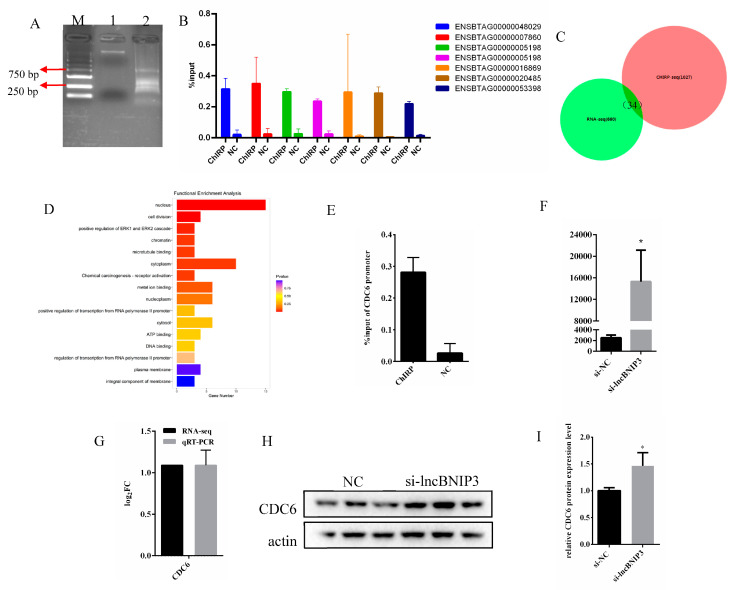
Interference with lncBNIP3 promoted the promoter activity and expression of CDC6. (**A**) Gel electrophoresis results of fragmentation products in the ChIRP test (M for Marker, from top to bottom are 5000 bp, 3000 bp, 2000 bp, 1500 bp, 1000 bp, 750 bp, 500 bp, 250 bp, 100 bp; 1 for gDNA; 2 for the fragmentation products (**B**) RT-qPCR results of the expression levels of 7 peaks in ChIRP group and NC group relative to input group; (**C**) The intersection analysis between the genes with peaks of ChIRP-seq on promoters and DEGs of RNA-seq; (**D**) Functional enrichment analysis of the intersection genes; (**E**) RT-qPCR results of the peak on CDC6 gene promoter region in ChIRP and NC groups relative to input group; (**F**) The results of luciferase activity of CDC6 gene promoter after transfecting si-lncBNIP3 and si-NC into the bovine intramuscular preadipocytes; (**G**) The fold changes of CDC6 mRNA expression after transfection, with si-lncBNIP3 and si-NC measured by RNA-seq and RT-qPCR; (**H**) Western Blot results of CDC6; (**I**) Statistical data of the Western blot results. Data are expressed as means ± SD. * indicated significant difference (*p* < 0.05).

**Figure 7 ijms-24-04234-f007:**
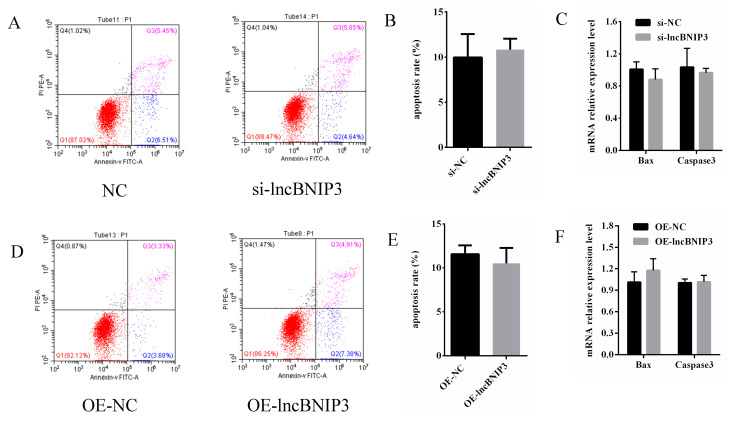
LncBNIP3 had no significant effect on the apoptosis of bovine intramuscular preadipocytes. (**A**) Flow cytometry results after transfecting si-lncBNIP3 and si-NC for 48 h; (**B**) Statistical data of the flow cytometry results; (**C**) Expression levels of Bax and Caspase3 mRNA after transfecting si-lncBNIP3 and si-NC for 48 h; (**D**) Flow cytometry results after transfecting OE-lncBNIP3 and OE-NC for 48 h; (**E**) Statistical data of the flow cytometry results; (**F**) Expression levels of Bax and Caspase3 mRNA after transfecting OE-lncBNIP3 and OE-NC for 48 h.

**Figure 8 ijms-24-04234-f008:**
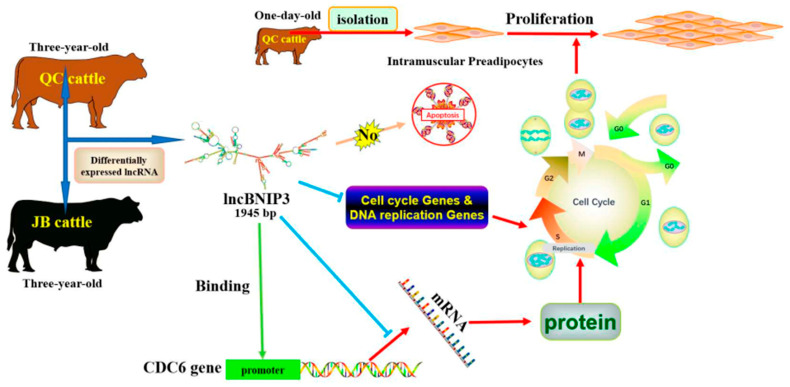
Model of the lncBNIP3 action in inhibiting the proliferation of bovine intramuscular preadipocytes. The green arrow indicate binding, the blue arrows indicate inhibition, and the red arrows indicate promotion.

## Data Availability

The data supporting the present study are available from the corresponding author upon reasonable request.

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
