# Peer review of "Long Non-Coding RNA BNIP3 Inhibited the Proliferation of Bovine Intramuscular Preadipocytes via Cell Cycle"

_ijms, 2023, doi:10.3390/ijms24044234_

Round 1

Author Response

Thank you for this valuable feedback.Those comments are very helpful for revising and improving our paper,as well as the important guiding significance to our researches. We have studied comments carefully and have made correction which we hope meet with approval. The main corrections in the paper and the responds to the reviewer’s comments are as flowing:

IJMS-2190200-PEER-REVIEW COMMENTS Title: Long non-coding RNA BNIP3 inhibited the proliferation of bovine intramuscular preadipocytes via cell cycle and CDC6

Abstract: You need more work in re-structuring the abstract to capture the real essence of the gene expression work.

Response: We are very much thankful for this important suggestion; the abstract has been revised as suggested “Previously, through a high-throughput sequencing analysis, we explored a 1949 bp lncRNA known as lncBNIP3. The 5’ RACE and 3’ RACE explored 1945 bp total length of lncBNIP3 including 1,621 bp of 5’RACE, and 464 bp of 3’RACE. The nucleoplasmic separation and FISH results explored nuclear localization of lncBNIP3. Moreover, the tissue expression of lncBNIP3 was highest in the longissimus dorsi muscle, followed by intramuscular fat. Furthermore, down-regulation of lncBNIP3 increased the EdU positive cell. The flow cytometry results showed that the number of cells in the S phase was significantly higher in preadipocytes transfected with si-lncBNIP3 than that of the control group. Similarly, CCK8 results showed that the number of cells after transfection of si-lncBNIP3 was significantly higher than that of si-NC. In addition, the mRNA expressions of proliferative marker genes CCNB1 and PCNA in the si-lncBNIP3 group were significantly higher than in the control group. The Western Blot (WB) results also showed that the protein expression level of PCNA transfection of si-lncBNIP3 was significantly higher than that of the control group. Similarly, the enrichment of lncBNIP3 significantly decrease the EdU positive cell in bovine preadipcytes. The results of flow cytometry and CCk8 assay also showed that overexpression of lncBNIP3 inhibited the proliferation of bovine preadipocytes. In addition, the overexpression of lncBNIP3 significantly inhibited the mRNA expressions of CCNB1 and PCNA. The WB results showed that overexpression of lncBNIP3 significantly inhibited the expression of CCNB1 at protein level. To further explore the mechanism of lncBNIP3 on the proliferation of intramuscular preadipocytes, RNA-seq was performed after interference with si-lncBNIP3, and a total of 660 differentially expressed genes (DEGs), including 417 up-regulated and 243 down-regulated were found. The KEGG pathway analysis showed that the cell cycle was the most significant pathway for the functional enrichment of DEGs, followed by the DNA replication pathway. The RT-qPCR verified the expression of twenty DEGs in the cell cycle. Therefore, we speculated that lncBNIP3 regulates intramuscular preadipocyte proliferation through the cell cycle and DNA replication pathways. To further confirm this hypothesis, intramuscular preadipocytes were added with cell cycle inhibitor Ara-C which inhibits DNA replication during the S phase of the cell cycle. Herein, Ara-C and si-lncBNIP3 were simultaneously added to the preadipocytes, and the CCK8, flow cytometry, and EdU assays were performed. The results showed that the si-lncBNIP3 could rescue the inhibitory effect of Ara-C in bovine preadipocyte proliferation. In addition, lncBNIP3 could bind to the promoter of CDC6 (Cell division control protein 6), and down-regulation of lncBNIP3 promoted the transcription activity and the expression of CDC6. Therefore, the inhibitory effect of lncBNIP3 on cell proliferation might be realized through the cell cycle pathway and CDC6. This study provided a valuable lncRNA with functional roles in intramuscular fat accumulation and revealed new strategies for improving beef quality.”

Always spell out any acronym in full at first mention, e.g., lncBNIP3, CDC6, and lncRNAs.

Response: We have spelled out any acronym in full at first mention.

No mention of the research and the hypothesis being tested.

Response: We have described the research and the hypothesis in the Introduction.

 A substantial part of the gene expression methodology is not mentioned in the abstract.

Response: We have added the gene expression methodology at your suggestion.

The authors need to insert a sentence or two on the RNA being extracted, cDNA synthesised, normalized and a standard curve plotted, etc.

Response: We have added these in Materials and Methods.

 L 20,182, 217, 281, 371, 384 And appears to be out of place here. Consider replacing it with another word or removing it. Suggestion: Use moreover, furthermore etc.

Response: We have revised these ‘And’ at your suggestion.   

Introduction:

L 28 A suitable reference(s) should be cited.

Response: We have cited a reference in L 28 at your suggestion.

L 28-30 Some of these studies should be cited here.

Response: We have cited two references in L 28-30 at your suggestion.

L 31 Remove the word ‘that’ as it is out of place.

Response: We have removed the word ‘that’ at your suggestion.

L 32 The phrase ‘most potent’ will fit in the sentence better than ‘strongest’.

Response: We have revised it at your suggestion.

L 38 The word ‘significant’ will help make the sentence sound clearer than ‘especially important’.

Response: We have revised it at your suggestion.

 L 64 The phrase ‘in contrast’ fits better than ‘while’. You may make the necessary adjustment.

Response: We have revised it at your suggestion.

 L 69-74 Appears like the conclusion of the study. This shouldn’t be stated or mentioned here. The authors should remove this aspect from here and summarise it and make it part of the conclusion.

Response: We have removed this aspect from here and summarized it and made it part of the conclusion at your suggestion. And we have added the research and the hypothesis of the study here. 

L 78 The article ‘the’ appears redundant, consider removing it.

Response: We have removed ‘the’ at your suggestion.

Materials and Methods

2.1. Animal Samples and cell culture

It should rather be ‘Animals, samples and cell culture’.

Response: We have revised it at your suggestion.

The authors didn’t highlight the condition under which the experimental animals were raised. Mention should be made of the feeding including the composition of the experimental diets, housing, and other environmental conditions. All these may have profound effects on the parameters of interest.

Response: We are very much thankful for this important suggestion; the following para has been added “In this study, three QC cattle and three JB cattle of the same age (3 years old) and gender (male), and the similar management conditions. The animals were fed a total mixed ration (TMR) containing 25% concentrate and 75% roughages of dry straw and corn silage, and water was offered ad libitum. The feeding was offered based on NRC standards (Nutrient Requirement of Beef Cattle) and in the similar rearing environment (similar temperature, humidity etc.). The animals were slaughtered and tissue samples including spleen, liver, lung, kidney, heart, subcutaneous fat, longissimus muscle and intramuscular fat were isolated.

2.3. Cell transfection After growing to 50-60% confluence, the intramuscular preadipocytes were transfected with siRNA (50 nM), OE-lncBNIP3 (1250 ng/mL) using Lipofectamine 3000 reagent (Thermo Fisher Scientific, USA) according to the manufacturer’s instructions, respectively. Comment: There is a need to write two or more sentences on this.

Response: We are very much thankful for this important suggestion; the following para has been added “Briefly, RNA oligonucleotides and the transfection reagent were separately diluted in Opti-MEM medium (Gibco) and incubated for 10 min at room temperature. Next, the two mixtures were combined and incubated for another 15 min at room temperature to allow for the formation of transfection reagent-RNA complexes. The transfection complexes were then added to the cell culture medium dropwise. The cells were incubated for 24 h before changing to fresh medium.”

2.5. Cytoplasmic and nuclear RNA separation

Cytoplasmic and nuclear RNA separation from intramuscular preadipocytes was performed using a PARISTM Kit (Life Technologies Corporation, Carlsbad, USA) according to the manufacturer’s instructions.

Comment: There is a need to write a few sentences on this as some of your readers may not be able to lay their hands on the protocol of the manufacturer.

Response: We have briefly described the manufacturer’s instructions at your suggestion. The following para has been added “Briefly, 10^7 cells were harvested and then incubated in the Cell Disruption Buffer for 10 min. The cytomembrane of the cells were cleaved. Subsequently, the cells were separated into cytoplasmic fraction and nuclear fraction. Finally, the RNA of cytoplasmic and nuclear fractions was isolated using Ambion RNAqueous® technology”

2.10. RNA Isolation, Library Construction, and Sequencing

It is necessary to highlight how RNA was extracted using Trizol, and how the concentration and integrity of the RNA were detected.

Response: (1) RNA Isolation was performed at following: Total RNA was extracted using Trizol reagent (Invitrogen, USA) with the manufacturer's instructions, according to the manufacturer’s instructions. Briefly, the cells were completely removed from the culture medium and rinsed three times with PBS. 1 mL of RNAiso was added to the cells in each 6-well plate and lysed on ice for 10 min. Then 0.2 mL of chloroform was added and shaken vigorously for 30 seconds. Subsequently, the mix above was put in a refrigerated centrifuge and centrifuged at 12000r/min for 15 min at 4 â—¦C. The supernatant was transferred to a new enzyme-free centrifuge tube and 0.5 mL of isopropyl alcohol was added. The new tube was centrifuged for 10 min after leaving, and then the supernatant was discarded and the total RNA was retained at the bottom. Finally, total RNA was washed using 75% ethanol and stored at -80 â—¦C.

(2) The RNA concentration and integrity were detected using a multifunctional luminometer (Tecan, Switzerland), and the OD260/280 values should range from 1.8 to 2.2 for subsequent research.

What were the PCR conditions used in this study?

Response: Quantitative Real Time (RT-PCR) was performed following the manufacturer’s protocol of Sybr Premix EX Taq Kit (Takara, Dalian, China) using thermocycler 7500 system SDS V 1.4.0 (Applied Biosystem, USA). Bovine β-Actin gene was used as an endogenous control. The thermocline conditions were: pre heating at 95 °C for 5 minutes, a total 34 cycles of denaturation at 95 °C for 30 seconds, annealing temperature at 60 °C for 30 seconds and extension temperature at 72 °C for 30 seconds. The relative mRNA expression levels were calculated using 2−ΔΔCt method.

Discussion

The results and data generated from this study are enormous, it is essential for the authors to look at it again and beef up certain areas.L 376 ‘in’ may be unnecessary in this sentence, consider removing it.

Response: We have removed it at your suggestion

L 379 Rephrase the first sentence. Suggestion: Our study found that…… L 386 Replace ‘ceases’ with ‘ceasing’.

Response: We have revised it at your suggestion

Conclusion

In keeping with the guidelines for IJMS, 395-404 should come under the heading conclusion.

Response: We have revised it at your suggestion

Reviewer 2 Report

1, What is the genetic background of these cattles? Is it random sampling? Are the two groups of cattle fed in the same farm? What cell line is used? More details are needed about these information.

2, The first figure in the Figure 1B has two bands. How to determine the destination band?

3, Is "positive cell" or "protive cell" in the Y-axis of Figure 2C and 3C?

4, Why is there CCNE1 only appeared in the mRNA relative expression level of overexpression part of the study(Figure 3G), but not elsewhere?

5, For the relative expression in protein levels of CCNB1 and PCNA(Figure 2I and 3I), please explain why interference with lncBNIP3 significantly affects PCNA, but significantly affects CCNB1 in the study of overexpression.

6, The research results (3.6 in Results) of CDC6 gene are more inference than verification. It is suggested to modify the article title or supplement the verification test.

Author Response

Thanks for this valuable feedback.Those comments are very helpful for revising and improving our paper,as well as the important guiding significance to our researches. We have studied comments carefully and have made correction which we hope meet with approval. The main corrections in the paper and the responds to the reviewer’s comments are as flowing:

1, What is the genetic background of these cattles? Is it random sampling? Are the two groups of cattle fed in the same farm? What cell line is used? More details are needed about these information.

Response: We are very much thankful for this important suggestion; the following para has been added in the manuscript please “In this study, three QC cattle and three JB cattle of the same age (3 years old) and gender (male), and the similar management conditions were selected from National Beef Cattle Improvement Research Center. The animals were fed a total mixed ration (TMR) containing 25% concentrate and 75% roughages of dry straw and corn silage, and water was offered ad libitum. The feeding was offered based on NRC standards (Nutrient Requirement of Beef Cattle) and in the similar rearing environment (similar temperature, humidity etc.). The animals were slaughtered and tissue samples including spleen, liver, lung, kidney, heart, subcutaneous fat, longissimus muscle and intramuscular fat were isolated. The isolation and culture of the bovine intramuscular preadipocytes was the same as previously described. The isolated cells were cultured in a 5% CO2 and 37°C incubator with DMM/F12 (Gibco, Grand Island, NY, USA) containing 10% FBS (Invitrogen, Invitrogen, US) and 1% antibiotics (100 IU/mL penicillin and 100μg/ mL streptomycin).

2, The first figure in the Figure 1B has two bands. How to determine the destination band?

Response: The two bands were inserted into the linearized pRACE vector. Subsequently, the two bands were sequenced and compared to known sequences. The successful alignment with the known sequence is the destination band.   

3, Is "positive cell" or "protive cell" in the Y-axis of Figure 2C and 3C?

Response: "positive cell" is correct in the Y-axis of Figure 2C and 3C, we have corrected it.  

4, Why is there CCNE1 only appeared in the mRNA relative expression level of overexpression part of the study(Figure 3G), but not elsewhere?

Response: It was an error on our part to measure the relative expression levels of the CCNE1 mRNA only for the overexpression part of the study, but not for the interference part. We have removed this part to be consistent with the interference part.

5, For the relative expression in protein levels of CCNB1 and PCNA (Figure 2I and 3I), please explain why interference with lncBNIP3 significantly affects PCNA, but significantly affects CCNB1 in the study of overexpression.

Response: To be honest, we were not sure of the exact reason. We speculated that overexpression of lncBNIP3 and interference with lncBNIP3 have different effects on the internal regulatory network, so the differentially expressed genes were not exactly the same after overexpression of lncBNIP3 and interference with lncBNIP3.

6, The research results (3.6 in Results) of CDC6 gene are more inference than verification. It is suggested to modify the article title or supplement the verification test.

Response: CDC6 is a crucial gene that affects the cell cycle and proliferation. Therefore, we did not directly explore the effect of CDC6 on intramuscular preadipocytes in our study. We have modified the article title at your suggestion.

Reviewer 3 Report

This manuscript investigated "Long non-coding RNA BNIP3 inhibited the proliferation of bovine intramuscular preadipocytes via cell cycle and CDC6". The content is fall into the scope of the present journal. The topic is interest, and the manuscript also raised many concerns. The follow are some specific comments.

Please rephrase the summary in accordance with whether the results of the study are significant or not, with the omission of the pronoun of the speaker.

Please delete the last paragraph in the introduction (Lines 69-74), as the introduction is not a place to display the results of the study

Did you test the normal distribution of the data before you decided to use ANOVA? I think that the data is not subject to a normal distribution due to the lack of sample numbers and then the high value of the standard deviation.

Figure 2-A, and Figure 2 and 3-F, Figure 6-F and I, please revise the statistical significant, the standard deviation is very high therefore I think that there was not a statistical significant

In Figure 2, 3 and 5-F No need to divide the vertical axis into two parts

The image resolution of Figure 1 to Figure 6 is not good, Please adjust it

Figure 5-E, I think that using the linear regression method will be the best choice for presenting the results

Figure 5-H, I think that using the two way anova will be the best choice for presenting the results The

Discussion needs to be further supported by the results of previous studies

Where is the conclusion of the study? Please include it under a separate heading

Author Response

Thank you for this valuable feedback.Those comments are very helpful for revising and improving our paper,as well as the important guiding significance to our researches.We have studied comments carefully and have made correction which we hope meet with approval. The main corrections in the paper and the responds to the reviewer’s comments are as flowing:

This manuscript investigated "Long non-coding RNA BNIP3 inhibited the proliferation of bovine intramuscular preadipocytes via cell cycle and CDC6". The content is fall into the scope of the present journal. The topic is interest, and the manuscript also raised many concerns. The follow are some specific comments.

Please rephrase the summary in accordance with whether the results of the study are significant or not, with the omission of the pronoun of the speaker.

Response: We are very much thankful for this important suggestion, and we have resived it at your suggestion.

Please delete the last paragraph in the introduction (Lines 69-74), as the introduction is not a place to display the results of the study

Response: We have deleted the last paragraph in the introduction at your suggestion.

Did you test the normal distribution of the data before you decided to use ANOVA? I think that the data is not subject to a normal distribution due to the lack of sample numbers and then the high value of the standard deviation.  Figure 2-A, and Figure 2 and 3-F, Figure 6-F and I, please revise the statistical significant, the standard deviation is very high therefore I think that there was not a statistical significant.

Response: We used the Shapiro-Wilk test of SPSS to do test the normal distribution of the data before we dicided to use ANOVA. Only the data conforming to normal distribution was analyzed by subsequent t-test. During the process of the t-test analysis, the Levene’s test was firstly performed to identify the homogeneity of variance. Then, the statistical significance was analyzed. According to your comments, we have re-analyzed the data in Figure 2-A, Figure 2-F, Figure 3-F, Figure 6-F and Figure 6-I, and these are all statistical significant.

In Figure 2, 3 and 5-F No need to divide the vertical axis into two parts.

Response: We have changed them at your suggestion.

The image resolution of Figure 1 to Figure 6 is not good, Please adjust it.

Response: The resolution of theoriginal images we submitted was good. However, the journal staff compressed the images during the manuscript type-setting, resulting in poor image resolution. We have adjusted it at your suggestion.

Figure 5-E, I think that using the linear regression method will be the best choice for presenting the results

Response: Thanks for your suggestion. Linear regression is a statistical analysis method that uses regression analysis in mathematical statistics to determine the quantitative relationships between two or more variables. In this part, the aim was to explore the optimal experimental concentration of Ara-C, rather than the quantitative relationship between concentration of Ara-C and absorbance at 450 nm, so we didn’t choose a linear regression method.

Figure 5-H, I think that using the two way anova will be the best choice for presenting the results 

Response: We are very much thankful for this important suggestion. We have used the two way anova for presenting the results of Figure 5-H.

The Discussion needs to be further supported by the results of previous studies

Response: We have added some results of previous studies in the Discussion at your suggestion

Where is the conclusion of the study? Please include it under a separate heading

Response: We have added the conclusion at your suggestion.

Round 2

Reviewer 1 Report

IJMS 2190200 Peer review report

Long non-coding RNA BNIP3 inhibited the proliferation of bovine intramuscular preadipocytes via cell cycle.

The text is written using British spellings, endeavour to use it throughout the manuscript. Check lines 12, 70 and 328 and write flavor in British spelling.

Line 10 The word ‘contribute’ should be in the past tense, ‘contributed’.

Lines 20,29 It seems that ‘EdU positive’ is missing a hyphen. Consider adding it. Also, write it in full in Line 20 Since it is first mentioned there.

Lines 29, 385 ‘preadipcytes’ is misspelt, Check, and correct it.

Line 110 The adjective ‘different’ is modifying ‘expressed’ instead of a noun or pronoun. Use an adverb (suggestion: differently) to modify a verb, adjective or another adverb.

Line 114 The article ‘the’ seems to be missing before genetic; consider adding it.

Line 114 Catttle is misspelt, check and correct it.

Line 461 ‘The’ article seems to be missing. Add it before ‘bovine’.

Line 462 Pre heating is missing a hyphen; consider adding it.

Author Response

Long non-coding RNA BNIP3 inhibited the proliferation of bovine intramuscular preadipocytes via cell cycle.

1. The text is written using British spellings, endeavour to use it throughout the manuscript. Check lines 12, 70 and 328 and write flavor in British spelling.

Answer: Thanks for your valuable comments. We have replaced the corresponding ‘flavor’ in the manuscript with 'flavour'.

2. Line 10 The word ‘contribute’ should be in the past tense, ‘contributed’.

Answer: Thanks for your advice. We have revised the manuscript (L10).

3. Lines 20,29 It seems that ‘EdU positive’ is missing a hyphen. Consider adding it. Also, write it in full in Line 20 Since it is first mentioned there.

Answer: Thanks for your advice. We have revised the manuscript in L21 and L30.

4. Lines 29, 385 ‘preadipcytes’ is misspelt, Check, and correct it.

Answer: Thanks for your advice. We have replaced the corresponding ‘preadipcytes’ in the manuscript with ‘preadipocytes’.

5. Line 110 The adjective ‘different’ is modifying ‘expressed’ instead of a noun or pronoun. Use an adverb (suggestion: differently) to modify a verb, adjective or another adverb.

Answer: Thanks for your advice. We have replaced the corresponding different’ in the manuscript with ‘differentally’. (L99)

6. Line 114 The article ‘the’ seems to be missing before genetic; consider adding it.

Answer: Thanks for your advice. We have added ‘The’ to the front of ‘genetic’ (L103).

7. Line 114 Catttle is misspelt, check and correct it.

Answer: Thanks for your advice. We have replaced the corresponding ‘Catttle’ in the manuscript with ‘cattle’.

8. Line 461 ‘The’ article seems to be missing. Add it before ‘bovine’.

Answer: We have added ‘The’ to the front of ‘bovine’.

9. Line 462 Pre heating is missing a hyphen; consider adding it.

Answer: We have added a hyphen between Pre and heating.

Reviewer 2 Report

Null

Author Response

We sincerely appreciate your recognition. Based on your review, as well as the comments of other reviewers, we have made further revision to the manuscript. Thank you again for your efforts and attention to improve the quality of our manuscript.

Reviewer 3 Report

I think that graph pad prism can’t fit the figures 4-D and G please ensure

Author Response

I think that graph pad prism can’t fit the figures 4-D and G please ensure.

Answer: Thank you for your careful review. Figure4 showed the analysis results of RNA-seq, and the visualization of theses results were completed in the R language environment. We have supplemented this part of the method in the Materials and Methods (L481).